# Advances in Molecular Genetics and Genomics of African Rice (*Oryza glaberrima* Steud)

**DOI:** 10.3390/plants8100376

**Published:** 2019-09-26

**Authors:** Peterson W. Wambugu, Marie-Noelle Ndjiondjop, Robert Henry

**Affiliations:** 1Kenya Agricultural and Livestock Research Organization, Genetic Resources Research Institute, P.O. Box 30148 – 00100, Nairobi, Kenya; werupw@yahoo.com; 2M’bé Research Station, Africa Rice Center (AfricaRice), 01 B.P. 2551, Bouaké 01, Ivory Coast; m.ndjiondjop@cgiar.org; 3Queensland Alliance for Agriculture and Food Innovation, University of Queensland, Brisbane, QLD 4072, Australia

**Keywords:** African rice, climate change, genomic resources, genetic potential, genome sequencing, domestication, transcriptome and chloroplast

## Abstract

African rice (*Oryza glaberrima*) has a pool of genes for resistance to diverse biotic and abiotic stresses, making it an important genetic resource for rice improvement. African rice has potential for breeding for climate resilience and adapting rice cultivation to climate change. Over the last decade, there have been tremendous technological and analytical advances in genomics that have dramatically altered the landscape of rice research. Here we review the remarkable advances in knowledge that have been witnessed in the last few years in the area of genetics and genomics of African rice. Advances in cheap DNA sequencing technologies have fuelled development of numerous genomic and transcriptomic resources. Genomics has been pivotal in elucidating the genetic architecture of important traits thereby providing a basis for unlocking important trait variation. Whole genome re-sequencing studies have provided great insights on the domestication process, though key studies continue giving conflicting conclusions and theories. However, the genomic resources of African rice appear to be under-utilized as there seems to be little evidence that these vast resources are being productively exploited for example in practical rice improvement programmes. Challenges in deploying African rice genetic resources in rice improvement and the genomics efforts made in addressing them are highlighted.

## 1. Background

African rice (*Oryza glaberrima* Steud) is one of the two rice species that have undergone independent domestication, the other one being Asian rice (*Oryza sativa*). African rice was domesticated about 3500 years ago from its putative progenitor, *Oryza barthii*. These two cultivated species play a vital role in enhancing food security in sub-Saharan Africa where the popularity of rice as a staple food is rising rapidly [1]. Despite this growing popularity of rice, the region is yet to attain self-sufficiency in rice production [1]. In order to achieve self-sufficiency, significant yield increases are required in order to ensure almost complete closure of existing gap between current and potential yields [2]. Climate change is however predicted to be a major threat that is likely to hamper the attainment of these enhanced yields in sub-Saharan Africa [3]. Being part of the *Oryza* primary gene pool and with its wide adaptive potential, African rice presents an important genetic resource that can support the breeding of high yielding climate resilient rice genotypes. Though its production is limited to only a few rice-growing agro-ecologies in West Africa, African rice is of global importance as it is a source of readily available genetic diversity for rice improvement. 

Genomic science presents novel tools for exploiting the genetic potential of African rice for accelerated rice productivity. Over the last one decade, there has been tremendous technological advances especially in DNA sequencing which have provided various genomic and genetic tools which have been pivotal in dramatically expanding the frontiers of crop research. Some of these advances include changes in sequencing instruments, chemistry, read length, throughput and bioinformatic tools. In African rice, some of the tools and resources that have been provided by these advances include complete genome reference sequences [4], novel mapping populations [5,6], bacterial artificial chromosome libraries [7] and numerous high throughput molecular markers [4,8]. Other advances include various analytical and bioinformatics tools, resources and platforms. This paper reviews some of the remarkable advances in knowledge that have been witnessed in the last few years in the area of genetics and genomics of this African indigenous *Oryza* species. Challenges in exploiting the immense genetic potential of African rice are highlighted.

## 2. Genetic Potential and Capacity for Climate Change Adaptation

Literature suggests that African rice possesses important traits that impart great adaptability to various biotic and abiotic stresses as well as climate change adaptation. The superior drought and thermal tolerance capacity of African rice has been reported [9]. This African indigenous rice species may have developed these traits as an adaptive mechanism against the harsh sahelo-saharan climate which is largely characterized by arid conditions. This drought tolerance is achieved through a series of morphological, phenological and physiological responses. Bimpong et al. [10] reported that, compared to Asian rice, some accessions of African rice have capacity to retain more transpirable water when faced with drought stress. These authors suggested that these accessions have capacity to close stomata early enough during periods of drought as a biological survival strategy that ensures effective use of available water. Some varieties of African rice are early maturing and are therefore able to escape terminal drought [11]. It has been found to have thin leaves that roll easily during drought thus reducing transpiration and thin roots which have a high soil penetrative capacity thereby helping in extracting water from the soil [12]. Its leaf and root architecture traits play an important role in enhancing drought tolerance. In a study conducted by Bimpong et al. [13], alien introgression lines derived from a cross between *O. glaberrima* and *O. sativa* had higher yields under drought conditions than the *O. sativa* parent. This demonstrates the potential of transferring drought related traits from *O. glaberrima* to *O. sativa*. About half of the beneficial alleles in the novel drought related quantitative trait loci (QTLs) identified in this study were derived from African rice. In a related study, Shaibu et al. [14] evaluated a total of about 2000 accessions of African rice for drought tolerance and found that some accessions had higher yields under drought conditions than the CG14 *O. glaberrima* drought tolerant check. Though the African rice genotypes were not significantly different from those of the *O. sativa* checks, they provide an important genetic resource for widening the gene pool that can be used to breed for drought tolerance. 

Natural variation that imparts greater thermal tolerance and adaptation to heat stress compared to *O. sativa* has been identified [9]. This adaptation is particularly important as recent modelling studies have reported potential massive rice yield declines in the West Africa’s Sahel region due to high temperature-induced reduction in photosynthesis [3]. African rice therefore possesses valuable genetic diversity for breeding for heat stress in the face of the ever-growing problem of climate change and variability. It has been found to be more tolerant to phosphorus deficiency than Asian rice [36]. Climate change is predicted to lead to increased soil salinity especially in low lying coastal areas and it is expected that this will cause a significant decline in rice yields [37]. Farmers in West Africa where salinity is high have reported that the key strategy they use in mitigating against salinity is planting of tolerant African rice varieties [8]. Owing to its high salt tolerance, African rice seems to be an important source of genes for breeding against salinity. Various types of predictions such as climate modelling show that of all regions, sub-Saharan Africa will be worst hit by climate change [38]. Incidentally, this region has low technological, financial and infrastructural climate change adaptive capacity. It is critically important that rice breeders in sub-Saharan Africa lay concrete strategies for exploiting these important African rice traits for climate change adaptation. Effective deployment of these adapted genetic resources will enhance the resilience and sustainability of rice production systems. In order to leverage the power of genomic tools in taking advantage of these traits, there is need to decipher the loci associated with this adaptive potential or phenotype. Table 1 summarizes the genetic potential of African rice in terms of resistance to a wide range of biotic and abiotic stresses among them drought, soil acidity, iron and aluminium toxicity and weed competitiveness [39]. 

## 3. Genetic and Molecular Basis of Important Traits

Genomic research holds the key to greater understanding and unlocking of genetic potential of both wild and domesticated species. In order to leverage the potential of African rice in rice improvement programmes there is need for sound understanding on the molecular underpinning of the functionally important variation. The lack of knowledge on the molecular and genetic basis of important traits acts as a major impediment in the deployment of African rice genetic resources in rice improvement. Aided by the increased availability of genomic resources and other remarkable advances in genomics and molecular genetics, the last couple of years have seen concerted efforts in linking genotypes and phenotypes. These have led to discovery of more loci or causal mutations associated with various traits particularly tolerance to various biotic and abiotic stresses. African rice has superior tolerance to a broad array of nutrient deficiencies and toxicities which are prevalent in most soils. In addition to previously detected QTLs for resistance to iron toxicity which seem stable across genetic backgrounds and environments, seven novel ones were identified [40]. Phosphorus deficiency is a major constraint in rice production particularly in sub-Saharan Africa. A novel allele that is associated with enhanced uptake of phosphorus has been identified in the *OsPSTOL1 (P-Starvation tolerance*) gene which is a major gene controlling the uptake of phosphorus. Candidate genomic regions that are associated with high mineral concentrations among them being key micronutrients have been identified [41]. A genome wide association study based on whole genome resequencing identified genomic regions controlling tolerance to salinity and geographic differentiation, with a total of 28 single nucleotide polymorphisms (SNPs) associated with various salt tolerance traits being identified [8]. This genetic resource of SNP markers is vital for plant breeding and adapting African rice to saline conditions.

Transcriptomic and histological analysis of African rice has identified a set of novel candidate genes for resistance to root knot nematode, *Meloidogyne graminicola*, a pest responsible for major yield losses in *O. sativa* [42]. A second major gene, *RYMV 2,* controlling resistance to *Rice Yellow Mottlel Virus (RYMV*) which is one of the most devastating rice infecting viruses in Africa, has been identified in *O. glaberrima* [26]. Efforts to fine map the *RYMV2* gene led to the identification of a putative loss-of-function one base deletion mutation in one of the candidate genes for *RYMV2*. This low frequency mutation was highly associated with *RYMV* resistance and affected a gene homologous to the *CPR5* defense gene in *Arabidopsis thaliana* [43]. Using *O. sativa* and *O. glaberrima* introgression lines, Gutierrez et al. [44] for the first time identified a major factor QTL controlling Rice stripe necrotic virus located on chromosome 11. These authors also identified a host of other QTLs for various traits, signifying the power of chromosome segment substitution lines (CSSL) as a genetic mapping tool. The continued identification of such locus and alleles is important in rice improvement as it assists in marker assisted selection.

The regulatory mechanisms of key domestication traits are increasingly being unravelled using genomics. Analysis of chromosome segment substitution lines with different genetic backgrounds revealed that the awnless phenotype in African rice was due to a novel recessive allele in the *Regulator of Awn Elongation 3 (RAE3)* gene located on chromosome 6 [45]. Other studies have identified genetic architecture of traits that were selected for by farmers during domestication for adapting the crop to their farming systems. A study by Li et al. [9] has uncovered the QTL responsible for thermal tolerance. Further analysis of this QTL identified a candidate gene, *OsPAB1 (Os03g0387100),* which was differentially expressed under heat stress and may have been selected for by farmers for adapting African rice to high temperatures. An African rice specific functional SNP, *H99*, in this gene was also identified that may allow the marker assisted introgression of thermal tolerance-enhancing alleles from African rice to other varieties. Despite the growing popularity of genome wide association studies, [8] its application seems limited in African rice as most researchers working on African rice seem to still rely on QTL mapping which has less resolution. Similarly, the application of systems genetics approaches to understand complex traits has been minimal or almost non-existent.

## 4. Genomic and Transcriptomic Resources

### 4.1. Genomic Sequences

African rice has one of the smallest genomes in the *Oryza* genus and its assembled reference is about 20% smaller than that of its domesticated counterpart (Table 2). Size differences between the various species are due to lineage-specific expansion and contraction of genes and gene families during the evolutionary process [46]. The first draft genome of African rice was presented by Sakai et al. [47]. This draft genome which was produced through whole genome shot gun sequencing had a size of about 206 Mb, which corresponds to about 0.6X coverage of the African rice genome whose size is estimated to be about 357Mb [48]. Though this genome sequence provided some useful insights on genomic evolution of African rice, it had limited utility as a large portion of the genome was missing. A few years later, a much-improved reference sequence in terms of assembly and annotation was released by Wang et al. [4] under the International *Oryza* Mapping and Alignment Project (IOMAP). Based on the estimated size of the *O. glaberrima* genome, this reference seems incomplete. Recent studies have also reported various assembly errors [49,50,51]. The CG14 reference sequence was assembled against the *O. sativa* Nipponbare reference sequence and may therefore have missed some *O. glaberrima* specific polymorphisms. The *PSTOL1* locus which is the major gene controlling the uptake of phosphorus was for example found to be missing from the assembled CG14 reference. The *PSTOL1* locus is located within Phosphorus uptake 1 (*Pup1*) which is the major QTL for phosphorus uptake. Aligning the *PSTOL1* locus with unplaced scaffolds revealed that it was present in an unanchored scaffold belonging to chromosome 12. Further analysis revealed that this particular loci and the adjacent sequence of a *Pup1* specific INDEL region spanning about 90 kb is absent in the Nipponbare reference, thus explaining the gap in the CG14 reference in this particular genomic region [51]. Though the identified assembly errors and gaps may hinder its effective utility in rice genetics and genomics, this reference sequence is arguably the most valuable genomic resource for African rice and has opened opportunities for detailed studies on this species. The CG14 reference is also relatively poorly annotated [52,53]. Owing to the importance of this species as a source of readily accessible diversity for rice improvement, there is need for concerted global efforts from the rice scientific community to initiate efforts aimed at improving the quality of this reference sequence.

Genomic studies have revealed that one reference sequence is not enough to represent the full genetic variability present in a species [57,58]. It is against this background that additional varieties of African rice were sequenced. Moreover, as stated, recent studies have identified various errors and gaps in the CG14 reference in addition to its relatively poor annotation [52,53]. These challenges have necessitated additional sequencing efforts to address them. In this regard, sequencing, de novo assembly and annotation of two additional genomes was undertaken. Similarly, using the same de novo approaches, the CG14 genome was also reassembled. These sequencing efforts yielded assemblies which, though smaller and more fragmented, produced better resolution in some loci such as *RYMV1* than the original CG14 reference. As shown in Table 3, they also predicted more protein coding genes than the IOMAP generated reference [50]. A similarly higher number of genes were reported by Zhang et al. [46]. A high-quality reference genome is fundamental for various genetic and genomic applications such as functional and comparative genomics. Lack of quality genomic resources has in some cases limited capacity to validate gene function thereby hindering the unlocking of novel trait variation. 

In addition to the whole genome sequences, advances in genomics have presented opportunities that have fuelled the development of other types of genomic resources, key among them being high throughput genetic markers. The IOMAP led initiative in which the CG14 variety reference genome was sequenced, generated the first large set of genomic data for African rice. Sequencing 20 diverse accessions of *O. glaberrima* identified a total of 4,447,424 SNPs [4]. Recently, Meyer, et al. [8] generated a genome wide SNP map that contained a total of 2.32 million SNPs by resequencing a total of 93 landraces. Molecular characterization of the *O. glaberrima* accessions conserved in AfricaRice genebank using diversity arrays technology (DArTseq) led to the identification of 3834 polymorphic SNPs [59]. Over 1.4 million Simple Sequence Repeats (SSR) have been identified in the African rice genome [46,47] providing a useful set of genetic markers. The lack of a dedicated set of high throughput markers that can study polymorphisms in interspecific crosses between Asian and African rice has been blamed for the limited exploitation of African rice genetic resources in interspecific breeding [60]. In order to address this gap, Pariasca-Tanaka et al. [60] developed a cost-effective high-throughput genotyping panel comprising of 2015 polymerase chain reaction (PCR)-based SNPs out of which 322 were polymorphic between the two species. These genomic resources provide versatile tools for dissecting the genetic basis of agriculturally important traits, for population genomic studies and other modern breeding applications. 

### 4.2. Chloroplast Genome Sequences

The first chloroplast genome sequences were published by Mariac et al. [61]. Additional chloroplast sequences including data for multiple accessions were later reported by [62] (Figure 1). These authors used a combination of both de novo and read mapping approaches in assembling the genomes. To date, a total of six African rice chloroplast genomes have been released, with the sizes ranging from 132,629–134,661 bp. The significant differences in the sizes of the various released genome sequences can be attributed to the protocol used in retrieving the chloroplast sequences and the assembly approaches used. However, the genome assembled by Mariac et al. [61], with a size of 132,629 bp, appears to be unusually small and to the best of our knowledge is the smallest of all the *Oryza* genomes that have so far been assembled. These genome sequences are providing a versatile tool for use in population genetics, phylogenetic and phylogeographic studies. Wambugu et al. [62] used chloroplast sequences to establish the phylogenetic relationships between African rice and other species constituting the *Oryza* primary gene pool.

### 4.3. Transcriptomic Resources

Transcriptome analysis has played an important role in supporting the assembly, annotation and analysis of the African rice genome and that of other species including *Oryza* wild species. Wang et al. [4] generated, to our knowledge, the largest multi tissue transcriptomic data for African rice. This RNA sequencing data was used to identify assembly gaps in the CG14 African rice reference. Sequence analysis identified seven genes that were missing in the reference, but RNA data indicated that they were transcribed clearly pointing to genome assembly gaps. In the same study, RNA sequence data was used to conduct comparative analysis of domestication genes in African rice and its progenitor. RNA sequence data was used to confirm the deletion of some genes in African rice among them the ortholog of the *O. sativa* shattering gene (*OsSh1*) which may have been lost during the process of evolution. While transcription for the *O. glaberrima* shattering gene (*OgSh4*) was detected in *O. barthii*, expression level for this gene was found to be limited or absent in African rice. This RNA sequencing analysis together with the analysis of mutation profiles led to the conclusion that African and Asian farmers may have targeted the same traits and genes but sometimes selected different mutations during the domestication process. A similar conclusion was made by Win et al. [63] who used gene expression analysis to unravel the genetic mechanism underlying loss of seed shattering in African rice. Zhang et al. [46] used transcriptome, EST and homology searches to validate predicted gene models during the annotation of de novo assembled *Oryza* genomes. Zhang et al. [46] used transcriptome, EST and homology searches to validate predicted gene models during the annotation of de novo assembled *Oryza* genomes. Further insights into the domestication process were given by Nabholz et al. [64] who used transcriptome sequencing to analyse the genetic diversity of various African rice transcripts. Genetic variation in African rice was reported to be the lowest for all grass species and perhaps for all domesticated crop species [64,65]. As noted by Ndjiondjop et al. [59], it might appear puzzling how a species with such narrow genetic base can possess such unique and exceptional genetic potential in terms of broad resistance to a variety of biotic and abiotic stresses. However, this situation does not seem to be unique as other studies have reported a negative correlation between neutral and functional diversity [66].

Analysis of the transcriptome has been used to decipher the genetic and molecular basis of important morphological, biochemical and physiological traits. African rice and *O. barthii* have uniquely different panicle architectures, but the underlying genetic cause has remained unknown. Comparative RNA analysis of these two African taxa revealed that these differences in panicle morphology are due to expression differences in the miR2118-triggered phased siRNAs [67]. RNA-seq analysis was used to elucidate the cytological and molecular mechanisms of resistance to *M. graminicola* root-knot nematodes, with differentially expressed genes being identified. [42]. Meyer et al. [8] used RNA analysis to identify genes that may be associated with tolerance to salinity based on their gene expression patterns. MicroRNAs are non-coding RNAs that may be involved in regulation of genes involved in response to various biotic and abiotic stresses. In a study analysing miRNAs that are involved in salinity stress response in African rice, Mondal et al. [52] identified a total of 150 conserved and 348 novel miRNAs which may have potential roles in gene expression. A total of 29 known and 32 novel differentially regulated miRNAs were identified suggesting they may have a direct role in response to salinity stress. Additional miRNAs belonging to different gene families have been reported for different *Oryza* species [46,68]. African rice has been found to have less polycistronic miRNA precursors compared to *O. barthii* [69], perhaps as a result of evolutionary and domestication processes. Identification of these important gene expression regulators and their analysis will aid in giving greater insights into their functional and evolutionary roles. This information on the genetic and molecular basis of various traits in African rice is useful to plant breeders as the loci identified can be genetically manipulated in order to impart increased tolerance to biotic and abiotic stresses. 

## 5. Supporting the Conservation and Utilization of African Rice Germplasm Using Genomics

African rice has huge genetic resources which are conserved in various ex situ conservation facilities globally. The largest collection totalling about 3910 accessions is held at AfricaRice Genebank, with the second largest collection of about 2828 accessions being conserved at the International Rice Research Institute (IRRI). As already stated, these collections are a rich reservoir of genes and alleles that is important for rice improvement particularly on tolerance to biotic and abiotic stresses. However, this diversity remains grossly underutilized [23,39]. Over the years, biotechnology-based approaches such as molecular markers have played a key role in genebank management. The current advances in genomics, particularly in DNA sequencing, are offering tools that have capacity for revolutionising the conservation and utilization of plant genetic resources [70,71]. However, compared to other areas of plant science, biodiversity conservation has been slow in embracing these technological advances [72]. Recently, there has been a commendable attempt towards leveraging these genomic-enabled advances in supporting the conservation and utilization of African rice germplasm currently conserved at the AfricaRice genebank. The molecular characterization of this collection using high density molecular markers has recently been reported. A total of 2927 accessions were genotyped with 31,739 DArTseq-based SNP markers. This data has assisted in identification of duplicates, constitution of core and mini-core collections as well as identifying human errors during various genebank operations [59,73]. SNP genotyping is assisting in revealing cases of taxonomic misidentification [74] which is common in genebanks and negatively impacts deployment of genetic resources in plant breeding and other research purposes. This is arguably the largest molecular data collected on this collection and presents a valuable resource for supporting decision making on key conservation aspects. Species SNP diagnostic markers which have capacity for accurately discriminating various *Oryza* species have been developed. Next generation sequencing based approaches have been used to identify duplicates in genebank collections [75] thereby providing a basis for rationalising germplasm collections. The current genomics-enhanced revolution will continue providing novel genomics, analytical and breeding tools that allow more rational and efficient conservation as well as more targeted exploitation of genetic resources. 

One of the greatest challenges limiting the use of genetic resources particularly those conserved in genebanks is inadequate understanding of their potential genetic value due to inadequate characterization [72,76]. Genetic diversity especially for genebank samples is usually studied anonymously with little or no efforts to identify the functional diversity [70]. Efforts have been made to analyse the genetic variation of *O. glaberrima* conserved at AfricaRice genebank anonymously using molecular markers [74,77,78]. Genome sequencing has been used to identify functional diversity related to different traits particularly on tolerance to biotic and abiotic stresses [8]. However, in most instances, functional diversity has been studied in only a few genotypes for a particular trait, leaving most of the African rice intraspecific variation largely unknown [23,79]. This limited characterization can largely be attributed to cost related considerations as this remains a major limiting factor. Even with the reduced sequencing and genotyping costs, many labs and researchers particularly in developing countries can still not afford to undertake genomic analysis of large sample sizes. Analysis of bulked samples is becoming a popular approach in genetic mapping and population genetic studies as it allows cost effective analysis of a large number of samples [80,81]. Pool sequencing and whole genome-based bulk segregant analysis are some of the commonly used cutting-edge approaches [82,83,84,85,86]. The lack of quality phenotypic data is increasingly emerging as a major bottleneck in establishing phenotype-to-genotype relationships. The on-going rapid advances in genomics seem to be outpacing capacity to undertake high throughput phenotypic analysis. This calls for an urgent need to invest in human resource capacity and physical infrastructure that will ensure enhanced phenotyping capabilities. Major initiatives aimed at exploiting the vast genetic potential of African rice through intra and inter-specific crossing are currently underway. These initiatives include, Rapid Alleles Mobilization (RAM) and Methodologies and new resources for genotyping and phenotyping (MENERGEP) of African rice species and their pathogens for developing strategic disease resistance breeding programs, both of which are being implemented by AfricaRice and other partners [59].

## 6. Grain Quality and Its Genetic Control

While priority has been placed on breeding for high yielding crops especially in developing countries, there is also need to ensure that these varieties deliver nutritional security which contributes to human health. Although African rice has potential to contribute genes for improving rice quality [39], this genetic potential has not been deployed in rice improvement and remains poorly studied. However, research interest in the physicochemical and functional properties of starch in African rice is growing [86,87,88,89,90,91,92,93]. Analysis of starch physicochemical properties has revealed that it has unique starch traits [92], a finding that could perhaps explain the renewed interest in starch traits in African rice. Generally, it has higher amylose content (AC) than Asian rice and could be a potential natural source of slowly digestible starch, traits that could confer it potential health benefits [92]. It has been found to have wider diversity of AC than earlier reported [89]. The health benefits of high amylose foods are increasingly being recognised, with such foods being associated with positive gastro-intestinal indices. African rice therefore has potential for use in the development of functional foods [94]. Analysis of *O. glaberrima* introgression lines has revealed that African rice is a novel genetic resource for addressing micro nutrient malnutrition through bio-fortification [41]. 

Deploying African rice genetic resources in breeding for healthier rice is however constrained by the poor understanding of molecular and genetic mechanisms underlying the unique starch traits, such as AC. Unlike in the case of Asian rice, lack of knowledge on marker-trait associations has hindered the use of marker-assisted selection. Recently, a whole genome based bulk segregant analysis conducted by Wambugu et al. [86] identified genetic markers that are putatively associated with AC. By sequencing bulks of interspecific progenies with low and high AC, this study identified a G/A SNP associated with the *Granule Bound Starch Synthesis (GBSS)* gene located on chromosome 6. Other putative AC associated SNPs were identified in genes encoding the *NAC* and *CCAAT-HAP5* transcription factors located on chromosome 1 and 11 respectively and which have previously been associated with starch biosynthesis. Analysis of natural variation in the *GBSS* locus identified several novel non-synonymous SNPs whose functional importance is still unknown. This study provides useful insights on the genetic control of AC, with the identified candidate genes being novel targets for manipulating AC in African rice. 

## 7. Challenges in Deploying African rice Genetic Diversity in Interspecific Breeding

One of the greatest challenges that have constrained the deployment of African rice diversity in rice breeding is strong and remnant sterility observed in interspecific crosses with Asian rice. This limits rice breeders from taking advantage of heterosis between the two cultivated species. Over the years, there has been intense research efforts on the sterility barriers between the two cultivated species [95,96,97]. Various approaches have been used in overcoming these barriers, with the first successful cross being achieved about 3 decades ago through the use of another culture and embryo rescue techniques [15]. The use of these conventional biotechnological approaches led to the development of New Rice for Africa (NERICA) varieties, which is arguably the most successful rice improvement program in sub-Saharan Africa. Research has identified a host of loci which are associated with reproductive barriers in cultivated rice [98]. Among these is the *S_1_* locus, which has a major effect on this interspecific sterility [99]. Despite the huge initial success that was achieved in generating interspecific crosses between *O. sativa* and *O. glaberrima*, the process is still fraught with technical challenges in addition to being tedious and time consuming. 

A variety of other methodological approaches have been developed and their effectiveness in addressing these sterility challenges tested. As reported by Lorieux et al. [98], a multi institutional collaborative effort has made efforts to address the challenge of sterility barriers by developing interspecific bridge lines. These are interspecific crosses between *O. sativa* and *O. glaberrima* and are developed through marker assisted selection of progenies that are homozygous for the *S_1_^s^* allele. Due to the large introgressions of *O. glaberrima* genome in these crosses and by significantly increasing fertility in subsequent crosses with diverse *O. sativa* lines, they ensure effective exploitation of useful *O. glaberrima* genes in conventional breeding programmes. Using mutagenesis, Koide et al. [95] isolated a mutant with an allele in the *S1* locus which is associated with increased fertility. Through this forward genetics approach, these authors were able to create a neutral allele which facilitates crossing these two cultivated species. Another closely related challenge is segregation distortion which has been reported in various genomic regions associated with a sterility locus such as the short arm of chromosome 6 where the *S1* locus is located [44]. Segregation distortion may affect the accuracy of QTL mapping as it may cause the effect of some QTLs to be overestimated. QTLs mapping in regions segregating in non-mendelian fashion should therefore be interpreted with caution.

## 8. Origin and Domestication of African Rice

Although there has been exceptional interest in studying the domestication and evolutionary history of African rice over the years, this remains unparalleled to that of Asian rice whose domestication is perhaps the most studied of all crop species. Several theories on the origin of African rice have been proposed but the debate rages on. An Asian origin of this species has previously been advanced but has been rejected [65]. Proposals of African rice having been domesticated from Asian rice in West Africa have been put forward but received very little support [100,101,102,103]. The dominant theory around which many studies and opinions appear to converge postulates that African rice was domesticated from *O. barthii* in West Africa. This has been supported by studies using gene sequence analysis [65], chloroplast genome based phylogenetic analysis [62] and population genomics [4]. Despite this general consensus, there has been an underlying complexity in understanding the exact location where domestication took place. Whole genome resequencing studies [4,104,105] have provided great insights on the domestication process and especially on the domestication centre but with key studies giving conflicting theories. Using a population genetics approach, Wang et al. [4] were the first authors to map the domestication centre of African rice using whole genome analysis. By resequencing 94 *O. barthii* and 20 *O. glaberrima* accessions as well as comparative genetic analysis of selected domestication genes, these authors mapped the actual domestication centre along the Niger River, consistent with original proposals from Porteres [106] and later supported by Li et al. [65]. Moreover, this resequencing study identified the specific *O. barthii* population from which African rice was domesticated. Recently, analysis of 246 whole genome sequences similarly mapped the Inner Niger Delta as the domestication centre [105]. These findings have however been disputed by Choi et al. [104] who analysed whole genome resequencing data from 286 African rice and *O. barthii* individuals. These authors proposed a non-centric origin of African rice instead of the single origin theory which has been proposed by many previous studies. Moreover, they reported that the progenitor population proposed by Wang et al. [4] lacked genetic differentiation from *O. glaberrima* and had greater resemblance to *O. glaberrima* than *O. barthii*. They therefore concluded that this population may have been misidentified or constitutes a feral weedy population. Rather than settling the debate on the origin of African rice as would have been expected, it appears the era of whole genome data is leading to greater controversies and conflicting theories. The different approaches used in the analysis of whole genome sequences data and the interpretation thereof may be the cause of these contradictory theories and conclusions.

Characterization of domestication genes has enabled deeper understanding of the domestication process of various species. The recently released *O. barthii* reference assembly [56] forms a valuable resource that will allow more insightful analysis of the evolution and domestication of African rice. Genomic analysis is increasing our understanding on the molecular basis of domestication of African rice. While significant progress has been made in the identification and in some cases cloning of domestication genes in Asian rice, relatively little is known about these genes in African rice [45,63,107,108]. The *O. barthii* reference assembly [56] will facilitate identification and analysis of orthologous loci between the domesticate and its progenitor and hence allow in-depth understanding of the target domestication genes. Analysis of selected domestication genes shows that ancient farmers in Africa and Asia targeted the same set of genes during domestication making it an independent and convergent evolution [4,105]. There is increasing evidence indicating that the genetic and molecular basis of the key domestication traits are in some cases conserved between African and Asian rice [4,109] though in other cases the genes and mutation profiles are different [45]. As highlighted earlier, this points to convergent evolution between the two species driven by human selection. The domestication process was associated with major shifts in various morphological traits, among them being grain size where humans showed a strong preference for big seeds. However, in African rice, the selection process appears not to have followed the dominant trend as far as grain size is concerned as the cultivated species typically has smaller seeds than its progenitor. The shift to small seeds has been attributed to a SNP mutation in the *GL4* gene that led to a stop codon. Interestingly, this mutation also led to loss of seed shattering [110]. Analysis of 93 diverse African rice landraces identified *SH3* as an additional gene controlling seed shattering which together with *SH4* led to multiple seed shattering phenotypes [111]. Using association analysis and positional cloning approach, a C/T SNP underlying the loss in seed shattering was identified [63]. The transition from the prostrate growth of *O. barthii* to erect growth in African rice has been attributed to a mutation in the promoter region of the *PROG7* (*PROSTRATE GROWTH 7*) gene which is located on chromosome 7 [107]. A 113kb deletion mutation in the *RICE PLANT ARCHITECTURE DOMESTICATION* (*RPAD*) locus on chromosome 7 has been reported as an additional genetic factor controlling plant architecture in both Asian and African rice [112]. This knowledge is important for plant improvement as important genetic variation can be introduced by targeting these genes and mutations. The domestication process may have been associated with loss of important diversity which may need to be introduced back in a well-targeted manner. 

## 9. Conclusions

In order to meet the food and nutritional requirements of the rapidly growing human population, there is an urgent need to increase per capita rice production. More innovations in rice breeding present an option for achieving the much-needed increases in rice productivity. This can be achieved by the development of super-varieties which have capacity to produce high yields per unit area under low water and nutrient input, in addition to being tolerant to diverse biotic and abiotic stresses. African rice offers a variety of these agronomically important traits. Production of such varieties will require sound knowledge in rice genetics and genomics. There is need to leverage the genomic capabilities that have been presented by cheap genome sequencing technologies to advance their contribution in rice improvement. African resource remains an untapped resource that can play a vital role in the development of novel gene pools. Additional efforts are required in the development of more structural and functional genomic resources. Identification of more functional genetic diversity is also of great value in these efforts. The on-going phenotypic and molecular characterization of African rice genetic resources is also critical in enhancing the utility of these resources in rice improvement. 

## Figures and Tables

**Figure 1 plants-08-00376-f001:**
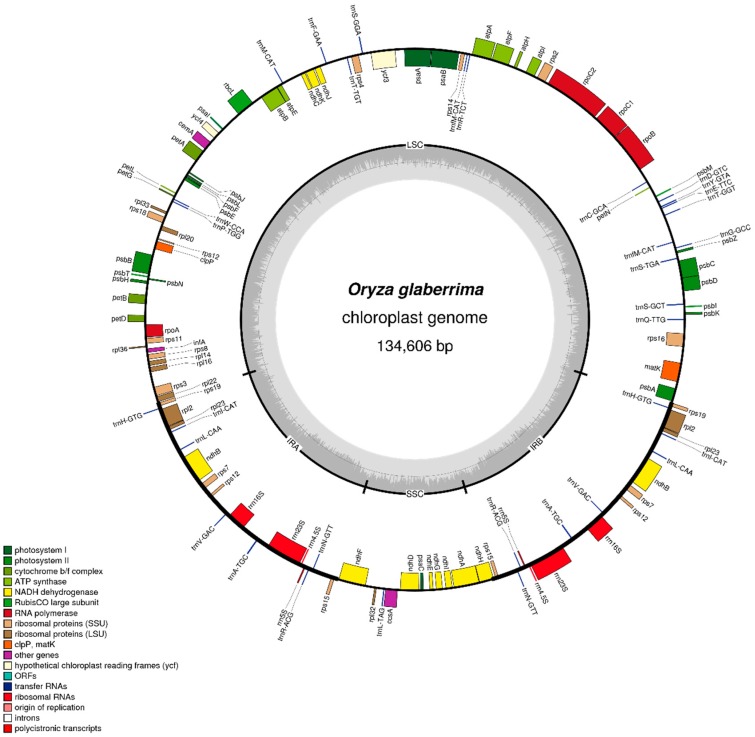
Gene map of *O. glaberrima* chloroplast genome [62].

**Table 1 plants-08-00376-t001:** Important traits on resistance/tolerance to biotic and abiotic stresses found in African rice.

Trait	Reference
Weed competitiveness	[15,16,17,18]
Drought tolerance	[15,19,20]
Resistance to nematodes	[21,22]
Resistance to iron toxicity	[23,24]
Resistance to African gall midge	[25]
Resistance to Rice Yellow Mortal Virus	[26,27,28,29]
Resistance to bacterial leaf blight (BLB)	[30,31]
Tolerance to lodging	[15,32]
Resistance to green rice leafhopper (*Nephotettix cincticeps* Uhler)	[31]
Tolerance to salinity	[8,33,34]
Tolerance to soils acidity	[19]
Tolerance to submergences	[35]

**Table 2 plants-08-00376-t002:** Important assembly and annotation features of selected *Oryza* species.

Species	Feature		Reference
	Genome Size	Gene Count	
*O. glaberrima*	316 Mb	33,164	[4]
*O. sativa*	370 Mb	37,544	[54]
*O. brachyantha*	261 Mb	32,038	[55]
*O. barthii*	308 Mb	34,575	[56]
*O. meridionalis*	336 Mb	29,308	[56]
*O. punctata*	394 Mb	31,679	[56]
*O. glumaepatula*	373 Mb	35,674	[56]

**Table 3 plants-08-00376-t003:** Description of various African rice genome assembles.

Feature	CG14 (I-OMAP)	CG14	TOG5681	G22
Assembly size	316 Mb	299 Mb	292 Mb	305 Mb
Gene count	33,164	50,000	51,262	49,662
Scaffold N50	217 kb	10 kb	13 kb	14 kb
Sequencing platform	Roche/454 GS-FLX Titanium Sequencing and Sanger	Illumina	Illumina	Illumina
Assembly approach	Reads aligned to *O. sativa* refseq	De novo	De novo	De novo

Source: [4,50].

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
