# Peer review of "Advances in Molecular Genetics and Genomics of African Rice (Oryza glaberrima Steud)"

_plants, 2019, doi:10.3390/plants8100376_

Round 1

Reviewer 1 Report

In the manuscript entitled Advances in Molecular Genetics and Genomics of  African Rice (Oryza glaberrima Steud) by Wambugu et al., the authors reviewed recent advances in the area of genomics and genetics of African rice, Oryza glaberima. This manuscript summarized the useful traits and their molecular basis, genome sequences, chloroplast sequences, and transcriptome data recently obtained in African rice. Then the manuscript reviewed genetic stocks, grain quality, breeding programs with some issues, and evolution of African rice.

This review has some merit of publication for readers to provide comprehensible summary of the recent progress in this research area. The manuscript is clearly written in regard to English writing.

One of my main concerns is the construction of chapters. In some paragraphs, the text does not correspond to the title of its section. For example, in lines 103-120, the authors mainly describing about developing SNP markers and issues of phenotyping, though its title is Genetic and molecular basis of important traits. These sentences should move out to somewhere else.

Another example is lines 208-309, lines 338-352, and 383-399. The authors are describing genetic diversity in these three sections. It looks duplicating. I feel that the authors should combine these sections for providing better understanding for readers.

In addition, the section title “Potential health benefits of African rice” does not show its content. It should be about the grain quality.

Another concern is in lines 219-239. In this section, the authors are describing chloroplast genomic sequencing. The authors compared 6 published sequences and mentioned that one of these six sequences has “possibly assembly error”. The authors also mentioned that “We recommend a review of this reference sequence.” As a scientist, I really understand what authors mean. However, I think this is out of scope of this review. The authors mentioned “Analysis of this genome has revealed that it is difficult to clearly delineate some of the distinct regions of the genome – long single copy, short single copy and the repeat regions – thereby pointing to the possibility of an assembly error.” without showing the result of “Analysis of this genome”. I think authors should show the result of “Analysis of this genome” and claim the genome’s review in somewhere else, not in the review paper.

Minor comments

Some sentences are in different size/color of letters. Lines 38-40. Lines 419-420. Lines 464-466.

Gene name should be italicized. For example, OsPAB1 (Os03g0387100), OsSh1, OsSh4, and PROG7.

Reviewer 2 Report

The review paper has been well organized and well written but needs to be revised as follows:

In line 21, why "re-sequencing" instead of sequencing has been used, can be explained. Table 1 can include some important findings. In Tables 2 and 3, gene count: 33,164 is a duplication and can be deleted. All tables should be well self explanatory. Line 221: reference in the text should be numbered. Fig. 1 is a duplication of a published paper and needs to be modified based on published papers or can be deleted. Careful revision is needed and all scientific names should be in italics.

Round 2

Reviewer 1 Report

The authors adequately answered to the most of reviewers' comments. However, I still have several minor comments need to check/modify.

Line 129, RYMV is for Rice Yellow Mottle Virus, not Rice Yellow Mortal Virus. Line 125, Please delete one of [8]s. Line 128, meloidogyne graminicola, please use capital for the initial M. Line 134, Arabidopsis thaliana should be italic letters. Line 143 Regulator of Awn Elongation 3 is RAE3, not RA3. Line 168 and other lines, authors are using both CG14 and CG 14. Better use one of the two. Line 205 3834 please use 3,834. (also for lines 222, 283, 284, 295)

Author Response

All minor corrections suggested by the reviewer have been made in the corrected manuscript.